# Beyond ECMO Survival: Long-Term Symptom Burden and Quality-of-Life Impairment in Hantavirus Cardiopulmonary Syndrome Survivors

**DOI:** 10.3390/v17091241

**Published:** 2025-09-15

**Authors:** Gonzalo Valenzuela, Katherine Barahona, Camila Rojas, Aldo Barrera, Carolina Henríquez, Constanza Martínez-Valdebenito, Marcela Potin, Paula Bedregal, Marcela Ferrés

**Affiliations:** 1Departamento de Enfermedades Infecciosas e Inmunología Pediátrica, Facultad de Medicina, Pontificia Universidad Católica de Chile, Santiago 8320165, Chile; ghvalenzuela@uc.cl (G.V.); avbarrera@uc.cl (A.B.); constanza.martinez.v@gmail.com (C.M.-V.); mpotin@ucchristus.cl (M.P.); 2Laboratorio de Infectología y Virología Molecular, Red Salud UC-CHRISTUS, Santiago 8320165, Chile; carohenriquezf@gmail.com; 3Servicio de Pediatría y Unidad de Paciente Crítico Pediátrico, Red Salud UC-CHRISTUS, Santiago 8320165, Chile; kntbarahona@gmail.com (K.B.); carojasr@ucchristus.cl (C.R.); 4Escuela de Salud Pública, Facultad de Medicina, Pontificia Universidad Católica de Chile, Santiago 8320165, Chile; pbedrega@gmail.com

**Keywords:** *Hantaviridae*, hantavirus cardiopulmonary syndrome, rodent-borne viruses, post-acute sequela, long-term symptoms, zoonosis, virus–host interaction, extracorporeal membrane oxygenation

## Abstract

Andes virus (ANDV) is the leading cause of hantavirus cardiopulmonary syndrome (HCPS) in South America, a severe zoonosis with high mortality. Advances in critical care and extracorporeal membrane oxygenation (ECMO) have significantly improved survival rates; however, data on recovery beyond survival remain limited. This multicenter cohort study enrolled laboratory-confirmed HCPS survivors in Chile between 2021 and 2024, with follow-up at 3–6 months post-symptom onset to assess physical and neuropsychological sequelae. Participants were stratified by ECMO requirement and the clinical severity of HCPS, and evaluated using self-reported recovery, standardized symptom questionnaires, and EQ-5D quality-of-life instruments. Among 21 survivors (11 ECMO, 10 non-ECMO), 61.9% reported incomplete recovery. While 60–70% of patients received general medical follow-up, only 30% of non-ECMO patients—compared to all ECMO patients—had contact with a rehabilitation provider. Motor dysfunction and palpitations were more frequent in ECMO survivors; however, Jaccard index analysis revealed clustering of physical and neuropsychological symptoms across both groups. EQ-5D assessments showed comparable quality-of-life impairment, though non-ECMO survivors more often reported pain/discomfort (90.0% vs. 63.6%) and higher rates of analgesic self-medication. These findings highlight the burden of persistent symptoms after HCPS and the need for multidisciplinary post-discharge care in endemic regions.

## 1. Introduction

Hantaviruses are rodent-borne viruses that cause two major clinical syndromes in humans: hemorrhagic fever with renal syndrome (HFRS) in Eurasia and HCPS in the Americas [1,2]. In South America, HCPS is caused by several hantaviruses, with ANDV being the primary etiological agent in Chile [1,3]. This single-stranded RNA virus, belonging to the family *Hantaviridae*, is transmitted to humans primarily via inhalation of aerosols contaminated with excreta from infected *Oligoryzomys longicaudatus* rodents, which are widespread from the Coquimbo region (30°56′ S) to as far south as the Aysén region (46°46′ S), in southern Chile [1,3,4].

The clinical course of ANDV-associated HCPS is unpredictable and characterized by abrupt capillary leak, hypoxemic respiratory failure, and shock [1]. For suspected or confirmed HCPS, patients with respiratory distress should be transferred early to an ICU, preferably a referral center with extracorporeal membrane oxygenation (ECMO) availability [1,5]. Management is primarily supportive, with careful use of fluids, early vasoactive therapy, and timely invasive mechanical ventilation; in cases with refractory hypoxemia or shock, ECMO has been associated with improved survival [1,5,6,7]. All these resources used at the right time can make a difference in the survival of previously healthy adults and children who turned out to be accidental hosts of this zoonosis.

Although the acute manifestations of ANDV infection are well characterized, and considerable efforts have been made to bring these patients to the forefront, its long-term effects remain poorly understood. Studies from Europe and Asia suggest that survivors of HFRS, particularly those infected with Puumala virus, may experience symptoms such as persistent fatigue, neurocognitive disturbances, and cardiovascular or endocrine sequelae [8,9]. However, whether similar post-acute consequences occur after HCPS remains unclear, and available data are limited to isolated case series [1,10,11,12]. Moreover, it remains challenging to distinguish the long-term effects of the ANDV or Sin Nombre virus infection from those related to all critical care and ECMO support [1,12].

The increasing recognition of post-viral syndromes, such as long COVID, has emphasized the importance of characterizing functional outcomes across infectious diseases. Validated tools, such as the EQ-5D, now enable standardized assessment of health-related quality of life in survivors of severe infections [13,14]. However, data specific to ANDV remain scarce, limiting the development of evidence-based follow-up strategies and rehabilitation programs in endemic regions [1,15,16].

This study sought to characterize the long-term clinical and functional consequences of ANDV infection in a cohort of HCPS survivors. By comparing ECMO and non-ECMO cases, as well as HCPS severity, we aimed to identify gaps in recovery and highlight the need for integrated post-critical care follow-up in endemic areas of hantavirus infection. Our findings provide novel insight into the long-term burden of HCPS and underscore the need for structured post-discharge care for all survivors of hantavirus infection.

## 2. Materials and Methods

### 2.1. Study Design and Setting

Between January 2021 and December 2024, we conducted a multicenter observational follow-up study of patients with laboratory-confirmed ANDV infection in Chile. Participants were identified from referral hospitals, which provided epidemiological and clinical data collected during hospitalization. Post-discharge data collection and symptom characterization were performed using standardized forms based on the ISARIC-WHO Clinical Characterization Protocol for severe emerging infections [17,18]. The study protocol was approved by the Ethics Committee of the Pontificia Universidad Católica de Chile (approval ID: 230324007). Informed consent for participation was obtained from all subjects involved in the study. This manuscript follows the Strengthening the Reporting of Observational Studies in Epidemiology (STROBE) guidelines [19].

### 2.2. Participants

Eligible participants included pediatric patients and adults with RT-qPCR-confirmed ANDV infection who had been hospitalized and admitted to the intensive care unit (ICU) during the study period. Laboratory confirmation was performed using RT-qPCR for ANDV on blood samples, rather than respiratory swabs, as this method is the test of choice for early and accurate diagnosis, demonstrating sensitivity of ~95% and specificity of 100% in peripheral bloods [20,21]. Informed consent was obtained for telematic follow-up. Convalescent post-discharge assessments were conducted via telemedicine or structured self-report surveys between three and six months after symptom onset. Patients diagnosed in 2021 and 2022 were retrospectively contacted and consented to report their post-infection symptoms and recovery status. No standardized post-ANDV rehabilitation protocol was in place during the study period.

### 2.3. Variables

Data collection instruments were adapted from validated post-acute COVID-19 and critical care follow-up tools, including those addressing post-acute sequelae [17,18]. The adapted questionnaires were previously validated in Spanish and were administered through real-time telematic interviews conducted via Zoom. All responses and follow-up data were securely stored in REDCap, hosted at Pontificia Universidad Católica de Chile [17,18]. Surveys took approximately 30 min to complete and were administered to patients and/or their caregivers. Only one response was allowed per participant and/or caregiver.

Baseline demographic and clinical variables included age, sex, race and ethnicity [22], comorbidities, and smoking status. Self-reported sedentary status before infection was assessed. A sedentary lifestyle was defined, for reference, as accumulating < 5000 steps/day, a criterion widely used in clinical and epidemiological research to denote low physical activity levels and elevated health risks [23]. Overweight and obesity were defined according to standard body mass index (BMI) thresholds. In adults, overweight was defined as a BMI ≥ 25 kg/m^2^ < 30 kg/m^2^, and obesity as a BMI ≥ 30 kg/m^2^ [24]. In children, overweight and obesity were defined according to the World Health Organization (WHO) Child Growth Standards. Overweight was classified as a body mass index (BMI) for age greater than +1 standard deviation (SD) and up to +2 SD from the WHO growth standard median, and obesity as BMI for age greater than +2 SD [25]. Malnutrition by excess encompasses both overweight and obesity. Support requirements included intensive care unit (ICU) admission, invasive mechanical ventilation, vasoactive agents, and ECMO use [1,5,6,7]. Patients with HCPS frequently develop shock, which was defined according to the PALISI (Pediatric Acute Lung Injury and Sepsis Investigators) consensus criteria. Shock was characterized by clinical signs of impaired perfusion (e.g., altered mental status, prolonged capillary refill, cold extremities), hypotension unresponsive to fluid resuscitation, or the need for vasoactive agents to maintain age-appropriate blood pressure [7,26]. Clinical HCPS severity was categorized as mild in cases presenting with fever and respiratory symptoms, with or without the need for supplemental oxygen delivered by mask, but without requiring mechanical ventilation. In contrast, cases were classified as severe hantavirus cardiopulmonary syndrome when patients required vasoactive support, mechanical ventilation, or ECMO [26].

### 2.4. Outcomes

Primary outcomes were self-reported recovery and the presence of persistent or new symptoms three to six months after symptom onset. Quality of life was assessed using the EuroQol EQ-5D-5L (for adults) and EQ-5D-Y (for pediatrics) instruments (registration number 75208), which evaluate five dimensions: mobility, self-care, usual activities, pain/discomfort, and anxiety/depression [13,14]. Participants rated each dimension both retrospectively (before infection) and at follow-up. Return to work or school, as well as changes in healthy lifestyle habits, were also recorded.

### 2.5. Statistical Analysis and Data Visualization

Categorical variables were summarized as frequencies and percentages, while continuous variables were reported as medians with interquartile ranges (IQR). Comparisons of categorical variables between groups were performed using Fisher’s exact test or Chi-squared test, as appropriate. The Mann–Whitney U test was used for comparisons of continuous variables between two groups. A *p*-value ≤ 0.05 was considered statistically significant.

All analyses and visualizations were performed using Python 3.13.5 (64-bit) in Visual Studio Code 1.101.1. The following libraries were used: pandas (v2.3) for data handling, SciPy (v1.16.0) for statistical testing and clustering, and seaborn (v0.13.2) and matplotlib (v3.10) for visualization.

Persistent symptoms were visualized using heatmaps stratified by ECMO status and HCPS severity, with symptoms ordered by overall frequency. Symptom co-occurrence was assessed via the Jaccard similarity index, and hierarchical clustering with average linkage was applied to generate a dendrogram. Radar plots were used to compare impairments in EQ-5D domains, and violin plots were generated to display the distribution and density of EQ-5D index scores across ECMO, non-ECMO, severe HCPS, and non-severe HCPS groups.

## 3. Results

### 3.1. Participant Characteristics at Admission and Clinical Hospitalization

Of the 36 eligible HCPS survivors with laboratory-confirmed ANDV infection identified between 2021 and 2024, 21 (65.6%) completed the follow-up assessment. Fifteen individuals were excluded from the study: three were initially reached, but communication was later lost and twelve could not be contacted. At hospital discharge, 17 (81.0%) of 21 survivors were classified as severe according to definition (Table 1). The remaining four (19.0%) were finally categorized as mild, although they were admitted in an ICU with ECMO as Chilean policy recommends.

Demographic and clinical characteristics of the ANDV survivors (n = 21) are summarized in Table 1. The cohort included seven individuals under 18 years of age (33.3%) and 14 adults (66.7%). Sex distribution was balanced, with no significant difference between groups. At diagnosis, more than half of the participants (57.1%) were classified as overweight, and 14.3% as obese. Reported comorbidities included smoking history (33.3%), sedentary lifestyle (33.3%), and mainly asthma (9.5%). No patients had a documented history of diabetes mellitus, cardiovascular disease, neurological disorders, or mood/anxiety conditions.

Sixteen (76.2%) survivors received invasive mechanical ventilation, with a median of 5.5 days; dysphagia was reported in three patients. None of the patients underwent tracheostomy. Eleven cases underwent ECMO support and five (45.5%) reported post-cannulation pain related to ECMO, and in two cases venous thrombosis was documented. No other hematologic disorders were identified. Fifteen survivors (71.4%) required vasoactive drugs during their hospital stay. Regarding antimicrobial therapy, 11 out of 21 survivors received β-lactams, 6 out of 21 were treated with vancomycin, and none received aminoglycosides.

The median length of hospitalization was significantly longer among ECMO survivors compared with non-ECMO patients (27 vs. 13 days, *p* < 0.01; Table 1). All patients experienced weight loss during their hospitalization. Median body weight reduction was 10 kg in ECMO survivors and 5.5 kg in non-ECMO individuals, corresponding to a median body mass index (BMI) reduction of 3.5 and 2.2 points, respectively.

No structured post-ICU rehabilitation program is currently available for hantavirus patients in Chile; consequently, none of the 21 HCPS survivors were enrolled in a formal multidisciplinary recovery pathway at discharge. Although 60–70% of patients received a general follow-up appointment with a physician, only three out of ten non-ECMO survivors (30.0% vs. 100%) were referred to specialized services, including dietetics, physical therapy, respiratory therapy, or mental health care. Notably, 7 of the 21 survivors (33.3%, all of whom were non-ECMO patients) received no support from any of these specialized providers. In contrast, all ECMO survivors were referred to at least one post-ICU service, reflecting a marked disparity in access to rehabilitation care. Furthermore, no patient, regardless of ECMO status, was referred to occupational therapy or follow-up by a nurse, highlighting essential gaps in the continuum of care.

### 3.2. Recovery and Persistent Symptoms at Convalescence Assessment

At the convalescent post-discharge assessment, 13/21 (61.9%) survivors self-reported incomplete recovery. This proportion did not differ by ECMO exposure (8/11 [72.7%] vs. 5/10 [50.0%]; Fisher’s exact *p* = 0.30). All participants endorsed ≥ 1 persistent symptom at follow-up, indicating a substantial burden of post-acute sequelae (Figure 1). Cumulative burden curves (percentage of patients by number of persistent symptoms) overlapped by ECMO status (Figure 1a), with no evidence of a difference (log-rank *p* = 0.729) and similar median counts with approximately 11–12 symptoms per patient at the 50th percentile. However, at higher thresholds, approximately 60% of ECMO survivors reported ≥10 symptoms, compared to 50% in the non-ECMO group. When stratified by clinical severity, curves were likewise not statistically different (Figure 1; log-rank *p* = 0.229). Nonetheless, the median symptom count appeared higher among severe cases (12 symptoms) than non-severe cases (6 symptoms), noting the small size of the non-severe group (n = 4). Collectively, these findings support the need for structured follow-up of all HCPS survivors, irrespective of ECMO use or initial severity. These findings underscore the importance of comprehensive follow-up for all HCPS survivors, irrespective of initial disease severity.

The most reported symptoms, affecting more than 50% of HCPS survivors at three months, were fatigue (81%), motor problems (71%), hair loss (67%), insomnia (62%), anxiety (57%), dyspnea (57%), memory problems (52%), sensory disturbances (52%), and nightmares (52%, Figure 2). A higher proportion of ECMO patients reported motor dysfunction (91% vs. 50%) and palpitations (55% vs. 10%) compared to non-ECMO patients, with a marginal statistical difference observed (*p* = 0.07 and 0.06, respectively). In contrast, non-ECMO survivors more often reported dyspnea (7/10, 70% vs. 5/11, 45%), anxiety (7/10, 70% vs. 5/11, 45%), cough (5/10, 50% vs. 3/11, 27%), low mood (6/10, 60% vs. 4/11, 36%), and early awakenings (5/10, 50% vs. 4/11, 36%). Vision problems were similar by ECMO status (2/11, 18% vs. 1/10, 10%; *p* = 1.00), whereas hearing problems occurred only in the non-ECMO group (3/10, 30% vs. 0/11, 0%; *p* = 0.09). In summary, these data indicate a substantial burden of post-acute sequelae across groups.

Hierarchical analysis of symptom co-occurrence using the Jaccard index revealed two phenotypic modules among HCPS survivors (Figure 3). First, a physical module showed the strongest pairwise co-occurrences observed: fatigue and motor problems (Jaccard index = 0.78), followed by motor and sensory problems (0.73), fatigue and hair loss (0.72), and fatigue and sensory problems (0.65). These combinations suggest a frequent overlap of fatigue, neuromuscular dysfunction, and somatic complaints. Second, a neuropsychiatric module also emerged, with anxiety and low mood exhibiting a Jaccard index of 0.57, fatigue and low mood (0.59), and insomnia with inattention and nightmares (all ~0.54). Additional moderate co-occurrences included memory problems with awakenings (0.54), and anxiety with sensory symptoms (0.53).

HCPS severity was mapped to the physical module in the dendrogram, positioned adjacent to fatigue, sensory disturbances, and motor problems. It showed co-occurrence with dyspnea and mood/sleep complaints, indicating that greater severity concentrates within the same cluster rather than forming a distinct module. ECMO status contributed two of the highest-ranked pairs (ECMO–motor problems 0.63; ECMO–palpitations 0.50), yet both the physical and neuropsychiatric modules were present across ECMO and non-ECMO survivors, suggesting that the co-occurrence architecture is not solely driven by advanced life-support exposure. However, both physical and neuropsychiatric modules were present across ECMO and non-ECMO groups, reinforcing the hypothesis that these symptom patterns may reflect the post-viral trajectory of HCPS rather than solely a consequence of critical illness severity.

### 3.3. Quality of Life and Functional Status at Follow-Up

In the overall cohort (n = 21), more than half of HCPS survivors report any problem (≥level 2 on the EQ-5D-5L) in pain/discomfort (76.2%), mobility (61.9%), and anxiety/depression (57.1%), highlighting a substantial burden of post-acute sequelae (Figure 4a); problems were less frequent in usual activities (47.6%) and self-care (19.0%). By ECMO status, pain/discomfort was more commonly reported in non-ECMO than ECMO survivors (90.0% vs. 7/11, 63.6%; *p* = 0.31). Among those with pain, moderate–severe intensity was documented in 6/9 (66.7%) non-ECMO and 2/7 (28.6%) ECMO survivors. In most cases, pain was primarily self-managed with analgesics.

Mobility impairment was similar between groups (7/11, 63.6% vs. 6/10, 60.0%; *p* > 0.99). Among affected individuals, moderate limitations were noted in 4/7 (57.1%) and 2/6 (33.3%), respectively. These individuals frequently described reduced walking capacity, difficulty developing work or sport activities, or muscle weakness, which hindered their return to routine daily activities. Anxiety/depression was reported by 6/11 (54.5%) ECMO and 6/10 (60.0%) non-ECMO participants (*p* > 0.99); among those affected, 60% rated symptoms as slight and 40% as moderate or severe. This symptom is frequently related to other symptoms such as insomnia, awakenings, inattention, nightmares, and/or memory problems.

When stratified by clinical severity, survivors with severe HCPS more often reported problems in mobility (11/17, 64.7%) and anxiety/depression (11/17, 64.7%) than those with non-severe disease (2/4, 50.0% and 1/4, 25.0%, respectively), with similar frequencies for pain/discomfort (13/17, 76.5% vs. 3/4, 75.0%); none of these differences reached statistical significance (all Fisher’s exact *p* ≥ 0.27), acknowledging the small non-severe stratum (Figure 4a). All domain results refer to any problem (slight to extreme) on the EQ-5D-5L.

Participants also self-rated overall health using the EQ-5D Visual Analogue Scale, a vertical scale anchored at 0 (“worst imaginable health”) and 100 (“best imaginable health”). On the survey day, the overall median EQ-5D VAS was 90 (IQR 60–90; range 40–100; n = 21), relative to the pre-ANDV infection baseline. EQ-VAS medians were similar by ECMO status (median 90 [IQR 60–90] vs. median 90 [IQR 60–90], *p* value = 0.97) with a Hodges–Lehmann median difference of 0 points, 95% CI −10 to 15 (Figure 4b). By clinical severity, medians were 90 (IQR 60–90; n = 17) for severe cases and 82.5 (IQR 67.5–90; range 60–90; n = 4) for non-severe cases (*p* = 0.81); median difference was 0 points, 95% CI −10 to 22). Estimates in the non-severe stratum are imprecise due to a small sample size. In summary, at three months of convalescence, self-perceived overall health was high at the median (EQ-VAS 90 [IQR 60–90]; range 40–100) but heterogeneous; there was no evidence of differences by ECMO status or HCPS severity, and a modest ceiling effect was observed (EQ-VAS = 100 in 9.5% of participants).

### 3.4. Social and Behavioral Impact

The return to regular activities was significantly impacted among HCPS survivors, with 19.0% of the patients unable to resume school or work six months after discharge. The median time to return to activity (work or school) was 3.5 months (n = 17) and did not vary by ECMO use (*p* = 0.36) or by HCPS severity (*p* = 0.24). A noticeable decline in academic or professional performance was self-reported, with similar proportions observed across ECMO status (63.6% vs. 50.0%; *p* = 0.67) and disease severity (58.8% vs. 50.0%; *p* = 1.00).

Social reintegration was similar in both groups, with 27.2% of patients in the ECMO group and 30.0% in the non-ECMO group reporting difficulties in socializing with peers. When stratified by severity, only survivors with severe HCPS reported socialization problems (29.4%); none were reported in the non-severe group. More critically, 45.5% (vs. 20%, *p* = 0.25) of the patients in the ECMO group perceived stigma at school or work, attributed to fears of rodent-borne contagion. The aforementioned stigma was also reported by 35.3% in severe HCPS survivors vs. 25.0% of mild HCPS patients (*p* = 0.74) These findings underscore the need for social support and understanding for HCPS survivors.

Significant changes in health behaviors were also observed among HCPS survivors. Self-medication for persistent symptoms was reported by 80.0% of non-ECMO survivors vs. 45.5% of ECMO survivors, and by 100% of survivors with mild HCPS vs. only 52.9% with severe disease at follow-up. The most used drugs included analgesics (such as acetaminophen, nonsteroidal anti-inflammatory drugs, and pregabalin), multivitamins, and sleep aids.

## 4. Discussion

Hantavirus cardiopulmonary syndrome (HCPS) is a severe zoonotic illness with high short-term mortality and substantial burden among survivors. While the acute phase of the disease has been extensively described in the literature, data on the long-term consequences in survivors remain scarce [1,9,10,11,12]. This multicenter study provides data on the ANDV post-acute functional, respiratory, and psychosocial consequences. These results demonstrate that a substantial proportion of patients, beyond the requirement for ECMO in the acute phase or severity of HCPS, experience persistent symptoms and impairment in their quality of life and social reintegration.

More than half of the participants reported feeling incompletely recovered three to six months after disease onset, indicating a high burden of persistent symptoms. These data are comparable to unvaccinated survivors of COVID-19, with no more than a 45% recovery rate [27]. ANDV infection is a rare zoonotic disease, and the number of survivors remains limited. From 2020 to 2025, Chile reported a median of 24 hantavirus survivors per year, corresponding to an estimated 60–75% of all laboratory-confirmed cases nationwide [1,28]. Our cohort highlights the need to identify and plan interventions for patients who have recovered from a severe and lethal disease. Notably, the use of validated instruments enabled a structured assessment of persistent symptoms at follow-up. Fatigue followed by dyspnea are the most frequent persistent symptoms in COVID-19 [29,30,31,32], but also in Puumala virus [8]. These findings were replicated in our cohort, with over 80% of survivors, regardless of ECMO requirement, reporting fatigue, and more than half of the patients experiencing dyspnea. This suggests that such symptoms are a frequent and expected consequence in survivors of a cytokine storm, which leads to endothelial injury and acute respiratory distress during the acute phase of the disease [31,32,33,34]. Moreover, we observed significant weight loss, with a decrease of 2 to 3 BMI points, resulting in noticeable emaciation and physical deconditioning [31,32,33,34]. Interestingly, this may also explain the presence of telogen effluvium, with hair loss observed in over 80% of patients, which is substantially higher than the rate reported following COVID-19 [35]. These findings support the need for structured physical and nutritional support upon hospital discharge to facilitate recovery and reintegration into daily activities.

Despite retinal hemorrhages and hearing loss having previously been reported in ANDV case series survivors [10], fewer than a quarter of the survivors in our cohort reported visual or hearing problems. These findings may be attributable to recent advances in intensive care medicine, including the implementation of more targeted therapeutic strategies such as optimized use of vasopressors and anticoagulation protocols. Hearing complaints after ICU care have been traditionally linked to exposure to aminoglycosides and loop diuretics, both recognized ototoxic agents [36,37]. In our cohort, none of the patients received aminoglycosides; however, more than 70% required vasoactive agents, which likely implied the frequent use of loop diuretics to mitigate pulmonary congestion. This raises the possibility of confounding by pharmacological exposure vs. the inflammatory effects of ANDV infection. We therefore recommend systematic documentation of ototoxic drug use and routine audiologic assessment during follow-up. HCPS induces a severe experience of dyspnea and imminent threat of death, which may constitute an acute stress reaction and potentially lead to post-traumatic stress disorder (PTSD) similar to Post-Intensive Care Syndrome [34,38,39]. Interestingly, this phenomenon was identified as a persistent symptom profile in HCPS survivors, even among those who did not require ECMO. This observation leads to the hypothesis of a possible viral origin, considering that hantavirus causes viremia and, consequently, multisystem involvement, rather than being solely attributable to disease severity. It is therefore essential to consider that many psychiatric conditions may have been underrecognized, given the limited access to mental health support following hospital discharge. These include a spectrum of disorders ranging from anxiety and mood disturbances, sleep disturbances to post-traumatic stress disorder, among others.

The profile of affected dimensions suggests that severe viral infections requiring ICU admission—such as COVID-19—compromise at least 50% of the five dimensions of the EQ-5D, highlighting a substantial burden across multiple domains of health-related quality of life [27,40]. In our cohort, the most critically ill group (ECMO) exhibited a clinical profile that was broadly comparable to that of non-ECMO survivors. Notably, pain-related problems were more prevalent among non-ECMO patients. This may reflect an underrecognized issue, as these individuals frequently relied on self-medication with analgesics and were not systematically referred for appropriate pain management. Post-ICU patients, regardless of the underlying etiology, typically report a 15–19% reduction in health-related quality of life at follow-up, as measured by the EQ-5D index [38]. In contrast, we show that survivors of HCPS caused by ANDV exhibited a decrease of only up to 10% in the EQ-5D index. This relative preservation of the EQ-5D score may reflect a limited self-perception of sequelae or a normalization of symptoms, particularly in the absence of well-characterized long-term complications associated with ANDV infection, given the lack of structured follow-up care.

The fact that less than half of HCPS survivors returned to work or school highlights significant challenges in social reintegration. In a prospective 1-year follow-up study, Myhren et al. reported that only 55% of previously employed or studying ICU survivors had returned to work or school, underscoring the profound and lasting impact of critical illness on functional and social recovery [38,40]. In addition, difficulties in socialization, experiences of stigma, and reported increases in the consumption of high-fat foods reflect broader disruptions in psychosocial and behavioral health. These findings underscore the urgent need to address recovery not only during the acute phase of critical illness but also throughout the convalescent period. Structured follow-up and timely referral to appropriate specialists, including mental health and nutritional support services, are essential to ensure comprehensive care and long-term recovery.

Although confirmation relied on RT-qPCR in blood, given its high diagnostic yield in ANDV, respiratory specimens may serve as complementary tools where feasible [20,21]. However, bronchoalveolar lavage is not routinely recommended due to its limited incremental yield and procedural risks [21]. To elucidate the mechanisms underlying long-term symptom persistence and sequelae after hantavirus infection, additional studies are needed to determine whether viral load in blood or respiratory samples correlate with long-term outcomes [1,20,21]. While most current investigations rely on blood samples for diagnostic confirmation due to their early and high sensitivity, it remains uncertain whether detection or higher viral burden in respiratory specimens contributes to the risk of long-term sequelae. Future research should aim to clarify these associations, as this could provide insights into viral pathogenesis and inform follow-up strategies in survivors [12].

One limitation of this study is the relatively small sample size. Current legislation in Chile restricts the possibility of conducting a nationwide study, as it requires that participants must have received care at one of the participating centers. As a result, our cohort includes only those patients who were either treated directly or referred to tertiary care centers with ECMO available. Concurrent stratification by severity and support modality reduces the overinterpretation of ECMO status contrasts, given confounding by indication [31,34]. In our cohort, EQ-5D problems and symptom clusters were broadly comparable across strata, albeit limited by small numbers, aligning with long-term trajectories reported after HCPS and other critical illnesses. A key limitation of the present study is the difficulty in determining whether the sequelae reported are directly attributable to hantavirus infection itself or are instead the result of critical illness in general. This challenge is also shared with other post-intensive care syndromes, including post-COVID-19 condition (also known as long COVID). Therefore, comparative research and mechanistic studies are needed to elucidate whether specific symptoms arise from direct viral pathogenesis rather than being solely related to post-ICU sequelae [1,20,21]. Despite these limitations, this study provides the first structured, multidimensional follow-up of HCPS survivors in Chile using validated tools. It highlights the need for systematic post-discharge monitoring protocols, especially given the significant functional impairments and socioeconomic repercussions observed.

## 5. Conclusions

These findings underscore the significant burden of post-acute symptoms and long-term functional limitations among survivors of HCPS. The persistent impairment across multiple health domains highlights the importance of implementing structured, multidisciplinary post-discharge care pathways. Targeted interventions—including physical, psychological, and social support—are particularly warranted for individuals residing in endemic regions, where access to specialized follow-up services may be limited. Our results advocate for national policies that prioritize long-term recovery and reintegration beyond the acute phase of care.

## Figures and Tables

**Figure 1 viruses-17-01241-f001:**
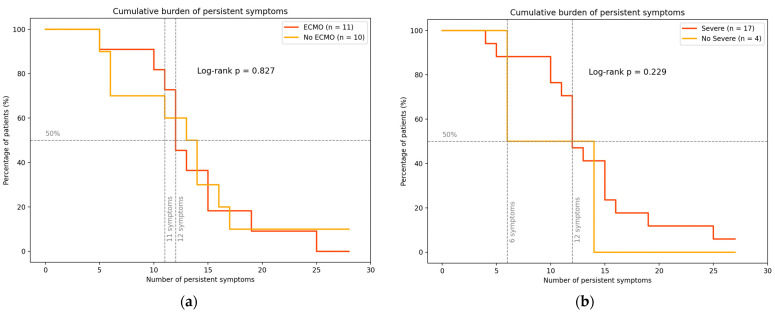
Cumulative distribution of persistent symptoms among survivors of hantavirus cardiopulmonary syndrome (HCPS), according to (**a**) ECMO requirement and (**b**) the clinical severity of HCPS. Cumulative proportions of patients reporting the number of symptoms are shown for ECMO (red) and non-ECMO (orange) groups. The curves intersect near the 50th percentile, suggesting a comparable median symptom burden between groups. These findings highlight the substantial and overlapping long-term sequelae in HCPS survivors, irrespective of initial disease severity or need for extracorporeal support.

**Figure 2 viruses-17-01241-f002:**
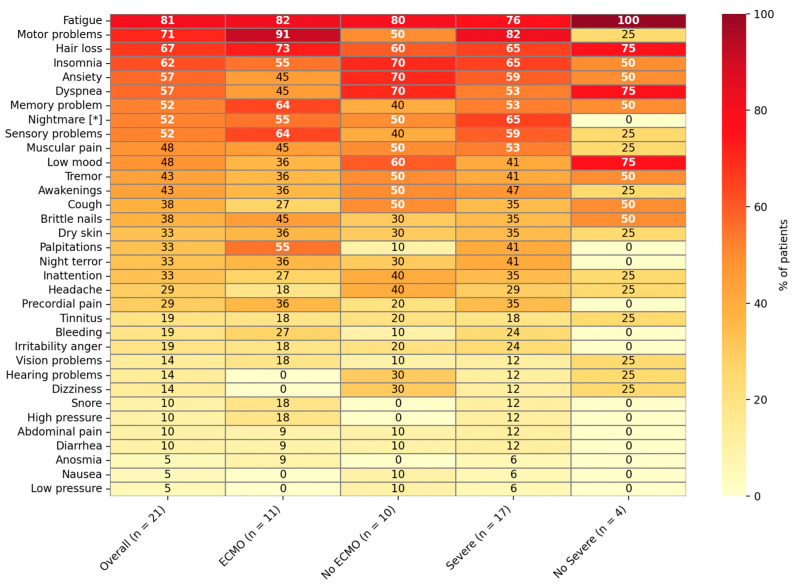
Persistent symptoms during the HCPS convalescence period, stratified by ECMO and HCPS severity, ordered by overall frequency. The heatmap shows the percentage of survivors reporting each symptom in the overall cohort (n = 21), the ECMO group (n = 11), the non-ECMO group (n = 10), the severe group (n = 17), and the non-severe group (n = 4). Rows are ordered by prevalence in the overall cohort; cell values are percentages, with warmer colors indicating higher prevalence (0–100%). Highly prevalent symptoms are indicated in red, while less frequent symptoms are indicated in yellow (e.g., fatigue, motor problems, hair loss, insomnia, anxiety, and dyspnea are among the most common, as shown in red). Asterisks (*) denote statistically significant differences between groups (*p*< 0.05, Fisher’s exact test). In this cohort, only nightmares met this criterion, reported in 11/17 (64.7%) severe survivors vs. 0/4 (0%) non-severe survivors (*p* = 0.035) *.

**Figure 3 viruses-17-01241-f003:**
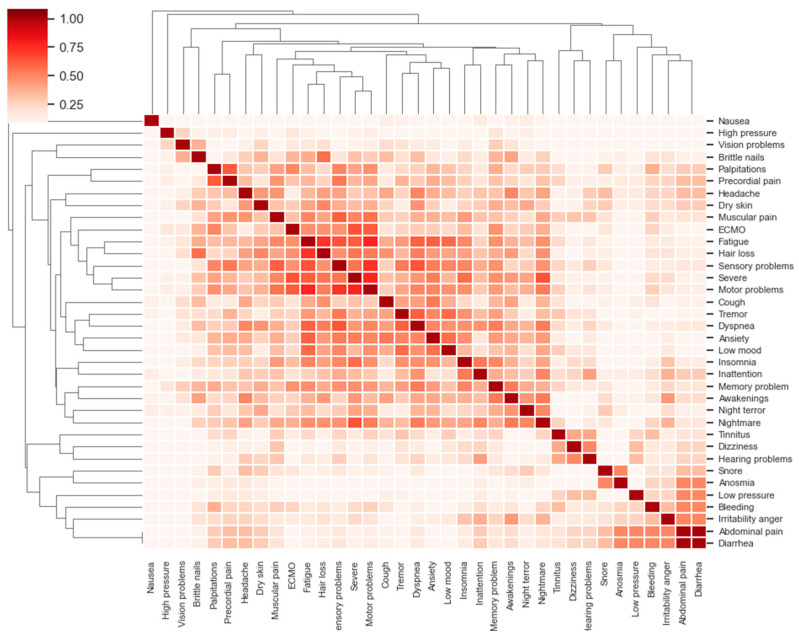
Persistent symptoms, ECMO requirement, and severe HCPS status co-occurrence matrix at the convalescent period post-HCPS using the Jaccard similarity index. The heatmap depicts pairwise co-occurrence of persistent symptoms reported by 21 survivors of Andes virus-associated HCPS, with hierarchical clustering applied to rows and columns. Color intensity represents the Jaccard index, ranging from 0 (no co-occurrence) to 1 (perfect co-occurrence).

**Figure 4 viruses-17-01241-f004:**
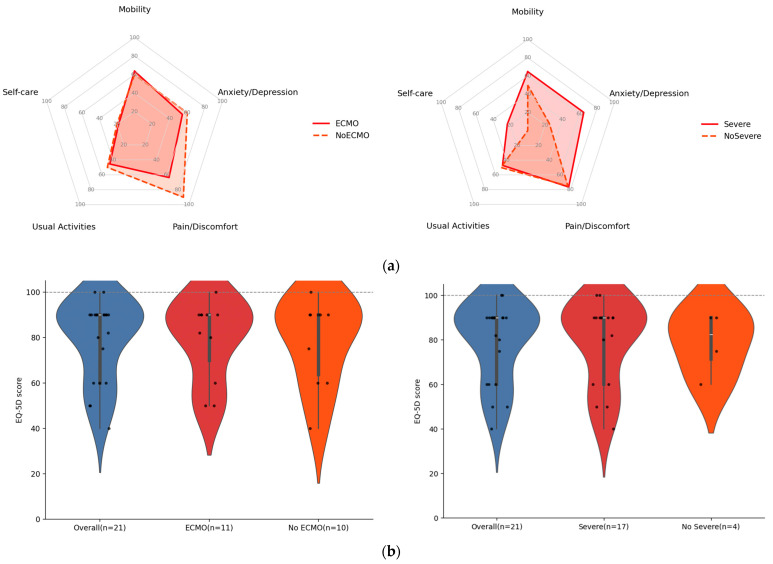
Health-related quality of life at convalescence assessed with EQ-5D. (**a**) Radar plots show the proportion of survivors reporting any problem in each EQ-5D dimension (Mobility, Self-care, Usual activities, Pain/Discomfort, Anxiety/Depression) stratified by ECMO use (ECMO, n = 11; noECMO, n = 10) and by HCPS severity (severe, n = 17; non-severe, n = 4). Dashed lines denote the non-ECMO or non-severe strata. Concentric rings represent increments of 20 percentage points. (**b**) Violin plots display EQ-VAS scores (0–100; higher scores indicate better health) for all participants (Overall) and stratified by ECMO status or HCPS severity. Each dot represents an individual survivor; thick lines indicate medians with interquartile ranges; the horizontal dashed line marks the anchor of 100 (best imaginable health).

**Table 1 viruses-17-01241-t001:** Demographic and clinical characteristics, along with discharge referrals, of hantavirus cardiopulmonary syndrome survivors, stratified by ECMO requirement.

Variable	Overall (n = 21)	ECMO, n (%) (n = 11)	No ECMO, n (%) (n = 10)
Demographic features			
Age, years, median (IQR)	30.2 (13.2–38.4)	31.3 (25.1–38.4)	20.7 (11.5–45.8)
Male, n (%)	11 (52.4)	5 (45.5)	6 (60.0)
Hispanic, n (%)	18 (85.7)	10 (90.9)	8 (80.0)
Comorbidities (e.g., obesity, asthma), n (%) ^a^	5 (23.8)	3 (27.3)	2 (20.0)
Malnutrition by excess, n (%) ^b^	12 (57.1)	7 (63.6)	5 (50.0)
Smoking history, n (%)	7 (33.3)	6 (54.1)	1 (10.0)
Sedentary lifestyle, n (%) ^c^	7 (33.3)	4 (36.4)	3 (30.0)
Acute clinical course			
ICU admission, n (%)	21 (100.0)	11 (100.0)	10 (100.0)
Invasive MV, n (%)	16 (76.2)	11 (100.0)	5 (50.0)
VAD, n (%)	15 (71.4)	11 (100.0)	5 (50.0)
ANDV immune plasma infusion, n (%)	17 (81.1)	11 (100.0)	6 (60.0)
Severity, n (%) ^d^ Mild, n (%) Severe, n (%)	4 (19.0) 17 (81.0)	0 (0) 11 (100.0)	4 (40.0) 6 (60.0)
BMI reduction, median (IQR)	2.3 (3.1–3.9)	3.6 (3.0–4.1)	2.3 (2.1–3.4)
Length of stay, days	16 (13.0–27.0)	27 (22.5–32.5)	13 (9.3–15.5)
Referral at discharge			
General practitioner/pediatrician, n (%)	14 (66.7)	7 (63.6)	7 (70.0)
Physical therapy, n (%)	11 (52.4)	8 (72.7)	3 (30.0)
Respiratory therapy, n (%)	5 (23.8)	4 (36.4)	1 (10.0)
Dietetics, n (%)	5 (23.8)	4 (36.4)	1 (10.0)
Mental health services, n (%)	4 (19.0)	3 (27.3)	0 (0)

Abbreviations: ICU: Intensive care unit; MV: Mechanical ventilation; VAD: Vasoactive drugs; ECMO: Extracorporeal membrane oxygenation; HCPS: Hantavirus cardiopulmonary syndrome; BMI: Body mass index; IQR: Interquartile range, presented as 25th to 75th percentiles. ^a^ Includes obesity, diabetes mellitus, chronic pulmonary disease, cardiovascular disease, neurological disorders, or mood/anxiety conditions. ^b^ Malnutrition by excess includes obesity and overweight. ^c^ A sedentary lifestyle is defined as accumulating fewer than 5000 steps per day. ^d^ Clinical severity was categorized as mild hantavirus infection in cases presenting with fever and respiratory symptoms, with or without the need for supplemental oxygen delivered by mask, but without requiring mechanical ventilation. In contrast, cases were classified as severe hantavirus cardiopulmonary syndrome when patients required vasoactive support, mechanical ventilation, or extracorporeal membrane oxygenation (ECMO).

## Data Availability

The original datasets generated and analyzed during the current study are not publicly available due to ethical and privacy restrictions but are available from the corresponding author on reasonable request.

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
