# Peer review of "Beyond ECMO Survival: Long-Term Symptom Burden and Quality-of-Life Impairment in Hantavirus Cardiopulmonary Syndrome Survivors"

_viruses, 2025, doi:10.3390/v17091241_

Round 1

Reviewer 1 Report

Comments and Suggestions for Authors

This study presents follow-up data on survivors of HCPS with a focus on both physical and neuropsychological sequelae in patients who required ECMO versus those managed with conventional critical care support. This study fills a gap in the literature and presents reasonable evidence of the potential long term sequalae from Andes HCPS. - Among the 21 survivors included in the study, ~62% reported incomplete recovery, highlighting the substantial long-term burden of HCPS regardless of acute phase interventions. It does warrant publication with some minor changes.

All patients required ICU but all but 4 must have received vasoactive support, mechanical ventilation, or ECMO (classified as severe HCPS), but it remains difficult to determine whether all cases truly represent equivalent severity or if variations in management thresholds (e.g., early ECMO vs aggressive ventilation) influenced outcomes, including the long term sequalae. i.e. are the 4 Milder cases of HCPS skewing results? The relatively small sample size (n=21) and the focus on severe ICU-managed cases limit generalizability but provide a valuable window into the high morbidity burden among survivors of critical HCPS.

Specific needs:

  • Would benefit by some analysis on the milder forms of HCPS – did they have similar sequelae?
  • IN Fig 2 there is missing info: There are no Bolded symptoms and no Asterix denoting stat significance – if there were none this should be stated.
  • In Fig 3 , I do not understand what the tree represents, especially the one on the left side of figure.
  • A more through discussion of the linkages between ototoxic agent during acute phase of disease

Authors should also consider discussing:

  • Expanding diagnostic sampling to include BAL alongside blood RT-PCR – or provide comment
  • Comparing long-term outcomes between mild, moderate, and severe HCPS cases, including those managed outside the ICU setting where feasible

Author Response

Comment 1.

“All patients required ICU but all but 4 must have received vasoactive support, mechanical ventilation, or ECMO (classified as severe HCPS), but it remains difficult to determine whether all cases truly represent equivalent severity or if variations in management thresholds (e.g., early ECMO vs aggressive ventilation) influenced outcomes, including the long term sequalae. i.e. are the 4 Milder cases of HCPS skewing results? The relatively small sample size (n=21) and the focus on severe ICU-managed cases limit generalizability but provide a valuable window into the high morbidity burden among survivors of critical HCPS.”

Response1. We agree that illness severity can confound long‑term outcomes. In Chile, in accordance with the national referral policy, all patients with a suspicion or diagnosis of Hantavirus are referred to an ICU center with ECMO availability. Therefore, our cohort comprises both severe cases and mild patients, as all patients were enrolled at the time of diagnosis, without knowing in advance how the disease would evolve.

To address this concern, we first stratified all analyses by clinical severity (mild vs. severe) in addition to ECMO vs. non-ECMO, and presented side-by-side summaries (Results 3.2–3.4; Figs. 1-4).

Clinical HCPS severity was categorized as mild in cases presenting with fever and respiratory symptoms, with or without the need for supplemental oxygen delivered by mask, but without requiring mechanical ventilation. In contrast, cases were classified as severe HCPS when patients required vasoactive support, mechanical ventilation, or ECMO.

In section, 3.2 Recovery and persistent symptoms at convalescence assessment. Cumulative burden curves (percentage of patients by number of persistent symptoms) showed that the median symptom count appeared higher among severe cases (12 symptoms) than mild cases (6 symptoms). The only symptom reaching statistical significance between severity strata was nightmares (64.7% in severe vs. 0% in mild; Fisher’s p = 0.035), although the small sample size (n = 4) should be noted. The symptom pattern, characterized by a high frequency of fatigue (100%), dyspnea (75%), hair loss (75%), and low mood (75%) in mild cases, suggests a persistent symptom profile regardless of severity.

In section, 3.3 Quality of Life and Functional Status at follow-up. Problems in mobility, pain/discomfort, and usual activities were similar in both groups (mild or severe cases, ECMO or non-ECMO groups).

In section, 3.4. Social and Behavioral Impact outcomes related to return to activities, academic or professional performance, social reintegration, stigma, and health behaviors were broadly similar when comparing survivors by ECMO use or HCPS severity, with no statistically significant differences observed.

Comment 2. ”Specific needs: Would benefit by some analysis on the milder forms of HCPS – did they have similar sequelae?”

Response 2. Thank you for this observation. We addressed the issue of milder HCPS forms in our response to Comment 1, where w

e clarified that survivors with mild disease showed outcomes broadly similar to those with severe HCPS or ECMO support.

Comment 3.Figure 2 lacks bolding for frequent symptoms and asterisks for significant differences; state explicitly if none.”

Response 3. Agree. Figure 2 now bolds ≥50% prevalence symptoms and uses asterisks for p < 0.05. The legend explicitly states that only nightmares met significance in this cohort.

Comment 3. “In Fig 3 , I do not understand what the tree represents, especially the one on the left side of figure.”

Response 3. We described Methods (2.5) to detail the pipeline (Jaccard similarity with average-linkage hierarchical clustering) and revised the Figure 3 legend to clarify that the dendrogram groups the observed data across survivors.

The left (as well as top, by symmetry) dendrogram illustrates the hierarchical clustering of symptoms based on Jaccard similarity. Branching patterns indicate how frequently symptoms co-occurred among survivors: those that clustered at lower levels of the tree were more closely associated. In comparison, those merging at higher levels showed weaker associations. Using the average linkage method, the algorithm calculated the mean pairwise Jaccard distances to iteratively group symptoms, thereby identifying natural clusters of post-HCPS sequelae.

This clustering approach has previously been applied to the analysis of long-term persistent symptoms, including post-acute sequelae of COVID-19. For instance, Sigfrid et al. employed similar methods to identify clusters of long COVID symptoms in adults discharged from UK hospitals, highlighting the utility of this approach in characterizing heterogeneous post-infectious outcomes.

References:

Sigfrid L, Drake TM, Pauley E, Jesudason EC, Olliaro P, Lim WS, Gillesen A, Berry C, Lowe DJ, McPeake J, et al. Long Covid in adults discharged from UK hospitals after Covid-19: A prospective, multicentre cohort study using the ISARIC WHO Clinical Characterisation Protocol. Lancet Reg Health Eur. 2021 Sep;8:100186. https://doi.org/10.1016/j.lanepe.2021.100186

Comment 4. A more through discussion of the linkages between ototoxic agent during acute phase of disease

Response 4. We added a new Discussion paragraph on potential ototoxicity from aminoglycosides and loop diuretics used in critical care, summarizing the mechanisms (outer hair cell injury, diuretic synergy, and genetic susceptibility) and emphasizing the need for structured medication capture and audiologic follow-up in future cohorts. Notably, hearing problems in our series were reported only in non‑ECMO survivors, suggesting possible confounding by drug exposures and/or disease effects.

Discussion added (line 1334-1341):

“Hearing complaints after ICU care have been traditionally linked to exposure to aminoglycosides and loop diuretics, both recognized ototoxic agents [36-37].  In our cohort, none of the patients received aminoglycosides. However, more than 70% required vasoactive agents, which likely implied the frequent use of loop diuretics to mitigate pulmonary congestion. This raises the possibility of confounding by pharmacological exposure versus the inflammatory effects of ANDV infection. We therefore recommend systematic documentation of ototoxic drug use and routine audiologic assessment during follow-up.”

Comment 5. Consider expanding diagnostic sampling to BAL alongside blood RT‑PCR—or comment.

We clarified in the Methods (2.2) and Discussion section that blood RT‑qPCR is the early diagnostic specimen of choice for ANDV (sensitivity ≈95%, specificity ≈100%). ANDV RNA has been detected in respiratory secretions during acute illness, supporting the use of complementary sampling. However, yields are variable, and routine BAL is not supported by evidence and may pose procedural/aerosol risks in unstable patients

Response 5.

Methods added (line 159-162)

“Laboratory confirmation was performed using RT‑qPCR for ANDV on blood samples, rather than respiratory swabs, as this method is the test of choice for early and accurate diagnosis, demonstrating sensitivity of ~95% and specificity of 100 % in peripheral blood [20].”

Discussion added (line 1379-1472)

“Although confirmation relied on RT-qPCR in blood, given its high diagnostic yield in ANDV, respiratory specimens may serve as complementary tools where feasible [20,26]. However, bronchoalveolar lavage is not routinely recommended due to its limited incremental yield and procedural risks [26]. To elucidate the mechanisms underlying long-term symptom persistence and sequelae after hantavirus infection, further studies are needed to determine whether viral load in blood or respiratory samples correlates with long-term outcomes [1,20,26]. While most current investigations rely on blood samples for diagnostic confirmation due to their early and high sensitivity, it remains uncertain whether detection or higher viral burden in respiratory specimens contributes to the risk of long-term sequelae. Future research should aim to clarify these associations, as this could provide insights into viral pathogenesis and inform follow-up strategies in survivors [12].”

Comment 6. Comparing long-term outcomes between mild, moderate, and severe HCPS cases, including those managed outside the ICU setting where feasible

Response 6. We addressed the issue of milder HCPS forms in our response to Comment 1, where we clarified that survivors with mild disease showed outcomes broadly similar to those with severe HCPS or ECMO support. Our classification did not include the moderate outcome; only severe and mild outcomes were considered.

Discussion added (line 1407-1411)

“Concurrent stratification by severity and support modality reduces the overinterpretation of ECMO status contrasts, given confounding by indication [31,34].  In our cohort, EQ-5D problems and symptom clusters were broadly comparable across strata, albeit limited by small numbers, aligning with long-term trajectories reported after HCPS and other critical illnesses.”

Reviewer 2 Report

Comments and Suggestions for Authors

The manuscript by Valenzuela et al. entitled “Beyond ECMO Survival: Long-Term Symptom Burden and Quality-of-Life Impairment in Hantavirus Cardiopulmonary Syndrome Survivors” summarized the burden of persistent symptoms of HCPS survival cases and this study will provide useful information for multidisciplinary post-discharge care in endemic regions. However, this reviewer raises some concerns as enumerated below:

Major concerns

In this study, as one of analyses, authors focused on comparison between ECMO supported and non-supported patients. However, this reviewer simply thinks that ECMO use is depended on disease severity of each patient, meaning that ongoing disease of ECMO patients were severe and those of non-ECMO patients were comparatively mild. Thus, authors should discuss the Long-Term Symptom Burden including original HPS severity in the cases in addition ECMO treatment. Although this study focus on survivors, were there significant differences of fatal (survival) rates between ECMO and non-ECMO patients? If those kinds of information are available, it is better to discuss about survivor’s information compared with fatal cases in both of ECMO and non-ECMO patients.

Minor concerns

Results section

It is better to describe summary of the results in each paragraph what authors claim those results in each paragraph for readers to understand.

Author Response

Comment 1. “Major concerns

In this study, as one of analyses, authors focused on comparison between ECMO supported and non-supported patients. However, this reviewer simply thinks that ECMO use is depended on disease severity of each patient, meaning that ongoing disease of ECMO patients were severe and those of non-ECMO patients were comparatively mild. Thus, authors should discuss the Long-Term Symptom Burden including original HPS severity in the cases in addition ECMO treatment. Although this study focus on survivors, were there significant differences of fatal (survival) rates between ECMO and non-ECMO patients? If those kinds of information are available, it is better to discuss about survivor’s information compared with fatal cases in both of ECMO and non-ECMO patients.”

Response1.

We agree that illness severity can confound long‑term outcomes. In accordance with the national referral policy, patients with suspected or diagnosed hantavirus infection are centralized to ECMO-capable intensive care units. Our cohort included severe and mild patients; both groups were enrolled a few days after diagnosis, without knowing the final disease outcome. Clinical HCPS severity was categorized as mild in cases presenting with fever and respiratory symptoms, with or without the need for supplemental oxygen delivered by mask, but without requiring mechanical ventilation. In contrast, cases were classified as severe HCPS when patients required vasoactive support, mechanical ventilation, or ECMO.

In section, 3.2 Recovery and persistent symptoms at convalescence assessment. Cumulative burden curves (percentage of patients by number of persistent symptoms) showed that the median symptom count appeared higher among severe cases (12 symptoms) than mild cases (6 symptoms). The only symptom reaching statistical significance between severity strata was nightmares (64.7% in severe vs. 0% in mild; Fisher’s p = 0.035), although the small sample size (n = 4) should be noted. The symptom pattern, characterized by a high frequency of fatigue (100%), dyspnea (75%), hair loss (75%), and low mood (75%) in mild cases, suggests a persistent symptom profile regardless of severity.

In section, 3.3 Quality of Life and Functional Status at follow-up. Problems in mobility, pain/discomfort, and usual activities were similar in both groups (mild or severe cases, ECMO or non-ECMO groups).

In section, 3.4. Social and Behavioral Impact outcomes related to return to activities, academic or professional performance, social reintegration, stigma, and health behaviors were broadly similar when comparing survivors by ECMO use or HCPS severity, with no statistically significant differences observed.

  1. Our cohort includes survivors only; therefore, mortality cannot be compared internally. For context, observational series of HCPS treated with ECMO report survival approaching two‑thirds in patients with otherwise prohibitive mortality, supporting ECMO as rescue therapy (non‑randomized studies).

Comment 2. Minor concerns. It is better to describe a summary of the results in each paragraph, as the authors claim, so that readers can understand.

Response 2.

We appreciate the reviewer’s suggestion. Summary statements have now been added at the end of each paragraph in the Results section (3.1–3.4) to highlight the authors’ claims and facilitate reader understanding explicitly.

Round 2

Reviewer 2 Report

Comments and Suggestions for Authors

Authors modified the manuscript properly following reviewer’s comments.